# Deciphering Factors Contributing to Cost-Effective Medicine Using Machine Learning

**DOI:** 10.3390/bioengineering11080818

**Published:** 2024-08-12

**Authors:** Bowen Long, Jinfeng Zhou, Fangya Tan, Srikar Bellur

**Affiliations:** 1Department of Analytics, Harrisburg University of Science and Technology, Harrisburg, PA 17101, USA; 2GM Financial, Fort Worth, TX 76102, USA

**Keywords:** cost-effective medicine, machine learning, cost-effectiveness rating (CER)

## Abstract

This study uses machine learning to identify critical factors influencing the cost-effectiveness of over-the-counter (OTC) medications. By developing a novel cost-effectiveness rating (CER) based on user ratings and prices, we analyzed data from Amazon. The findings indicate that Flexible Spending Account (FSA)/Health Savings Account (HSA) eligibility, symptom treatment range, safety warnings, special effects, active ingredients, and packaging size significantly impact cost-effectiveness across cold, allergy, digestion, and pain relief medications. Medications eligible for FSA or HSA funds, treating a broader range of symptoms, and having smaller packaging are perceived as more cost-effective. Cold medicines with safety warnings were cost-effective due to their lower average price and effective ingredients like phenylephrine and acetaminophen. Allergy medications with kid-friendly features showed higher cost-effectiveness, and ingredients like calcium, famotidine, and magnesium boosted the cost-effectiveness of digestion medicines. These insights help consumers make informed purchasing decisions and assist manufacturers and retailers in enhancing product competitiveness. Overall, this research supports better decision-making in the pharmaceutical industry by highlighting factors that drive cost-effective medication purchases.

## 1. Introduction

Over-the-counter (OTC) products are medications proven safe and effective for purchase without a prescription from a physician. They treat conditions such as pain, coughs, colds, diarrhea, heartburn, and allergies. They are readily available in pharmacies, grocery stores, gas stations, and online platforms. A survey conducted in March 2024 indicates that 93% of U.S. adults prefer using OTC medicines for minor health issues before seeking professional care, with 92% of physicians endorsing their effectiveness and safety [1].

With the rapid development of the Internet, there has been a significant opportunity for collaboration between the medicine retail industry and online platforms. The growing demand for home healthcare and wellness has fueled the expansion of online medicine purchases. Consumers increasingly rely on platforms like Amazon Pharmacy, Health Warehouse, and Optum, which are projected to account for about 35.26% of total revenue in the OTC pharmaceuticals market by 2024 [1].

Before deciding to buy OTC medicines, consumers go through several stages. These include recognizing a problem or symptom of a disease (problem recognition), finding appropriate information on drug indications (information search), evaluating alternatives (evaluation of other options), and finally, deciding on the proper medication (purchase decision). After purchasing, consumers evaluate whether the medication met their expectations and how satisfied they feel (post-purchase evaluation) [2,3]. Since decisions are primarily based on personal experience, they are subject to biases such as age, seriousness of the symptoms, and medication allergies, making the purchasing process complex. Therefore, consumers should pay more attention to information when searching and evaluating online medicine alternatives.

Amazon Pharmacy presents comprehensive prescriptions for colds, allergies, digestion issues, and pain relief medications. It is available for consultation 24/7. Each product listing on the Amazon website provides transparent pricing and user ratings, serving as indicators of perceived value that consider medications’ efficacy, quality, and user satisfaction. This information brings convenience and transparency to the online shopping experience. However, consumers are confronted with the daunting task of selecting from a vast array of medicines and their factors to address common conditions. Numerous factors related to OTC medicine impact customer decision-making, such as price, user reviews, efficacy, brand, size, ingredients, and side effects [4,5,6,7,8,9,10].

Price and perceived value influence consumer decisions [11,12,13,14]. Consumers tend to favor medications that offer higher value when prices are similar. Historically, consumers prioritized value over price, believing higher prices correlated with higher value (“you get what you pay for”). However, recent research suggests a shift in consumer behavior towards seeking cost-effective options. Gao et al. discovered that while price and perceived value are positively associated, higher prices do not always equate to proportionally higher perceived value [15]. Some consumers prioritize value over price, exhibiting lower price sensitivity. For instance, those seeking top-value products are less concerned with price and more willing to pay for perceived quality. At the same time, price-sensitive individuals may choose a lower-priced option even if it offers less perceived value than a higher-priced alternative.

Perceived value is the psychological balance consumers strike between expected gains and transaction sacrifices [16,17,18]. The correlation between value and price is a pivotal area of interest across industries, addressed prominently by Nelson’s “quality–price tradeoff” theory [19]. According to this theory, consumers weigh perceived value against price when purchasing, expecting higher value as prices increase. Creyer and Ross used a value index to show that consumers often opt for lower-priced, higher-value options over higher-priced, higher-quality ones [20]. Similarly, Zeithaml emphasized that product value—what consumers receive relative to what they pay—is critical in consumer decision-making [21]. Yoon et al. noted that shoppers use a value index (value = quality/price) to guide their purchasing decisions [22].

As medication costs rise significantly—some top-selling drugs have seen over 50% increases in fees since 2012 [23], with projections of further annual increases [24]—consumers face heightened pressure to balance value and price when selecting treatments. Considering this trend, choosing cost-effective products with competitive prices and a high perceived value is crucial. To assist consumers in making informed and cost-effective medication purchases, this paper first conducts web crawling to gather information on four common types of over-the-counter (OTC) medicines from Amazon Pharmacy: cold, allergy, digestion, and pain relief. It then introduces a novel cost-effectiveness rating (CER) indicator derived from a medicine’s user rating relative to its price. As a result of this CER, consumers are provided with valuable guidelines for navigating product choices based on price considerations.

Machine learning and deep learning are essential components of artificial intelligence, widely applied in healthcare, digital retailing, and social media [25,26,27,28]. In this study, our goal is to simplify customer decision-making when purchasing cost-effective medicines. To achieve this, we utilized machine learning models incorporating various variables such as medication ingredients, brand, manufacturer, and safety warnings extracted from Amazon web crawls to predict medicines’ cost-effectiveness ratings (CERs). We utilized various machine learning classifiers, such as logistic regression, decision tree, and multi-layer perceptron. Decision trees use nodes and branches to classify data based on attribute values [29], while random forests combine multiple decision trees to enhance accuracy [30]. XGBoost improves traditional gradient boosting with optimizations like parallel tree construction and pruning [31]. Linear discriminant analysis (LDA) reduces dimensionality and enhances class separability [32], and the K-nearest neighbors (KNN) method identifies nearby data points based on distance metrics [33]. MLP, a neural network with interconnected layers, learns complex data patterns for effective decision-making [34]. Finally, employing techniques such as SHAP values [35] and logistic regression, we explored the impact of each variable and identified key factors influencing medication cost-effectiveness.

Figure 1 illustrates the research framework of this paper. We employ machine learning models to predict the cost-effectiveness rating (CER) of over-the-counter (OTC) medications and enhance transparency with logistic regression and SHAP values to assess and identify key contributors to cost-effective medicines. Our contributions are as follows:
**Novel CER metric:** Introducing a metric that combines user ratings and prices to evaluate OTC medications, providing a comprehensive measure beneficial to consumers, manufacturers, and retailers.**Comprehensive factor analysis:** Evaluating factors such as FSA/HSA eligibility, symptom treatment range, safety warnings, special effects, active ingredients, and packaging size to identify key cost-effectiveness drivers across medication categories.**Explainable ML/AI techniques:** Assuring model accuracy and interpretability using SHAP values and logistic regression coefficients, aligned with the principles of explainable AI and the scope of the journal.**Cross-domain impact:** Bridging biomedical research and practical applications in the pharmaceutical industry by analyzing real-world data from Amazon, providing actionable insights for consumers, manufacturers, and retailers.
○**Consumers:** Simplifying the decision-making process, enabling informed choices about medication purchases, and balancing efficacy and cost, leading to enhanced satisfaction.○**Retailers:** Improving the shopping experience for retailers like Amazon Pharmacy by optimizing website layouts and recommendation systems [36], aiding in inventory management and targeted marketing strategies.○**Manufacturers:** Guiding product development by understanding consumer values such as efficacy, side effects, cost, and convencience [37], leading to more effective, user-friendly medicines, and better market positioning.

The structure of this paper is as follows: We present the dataset that describes the data processing and the applied machine learning models. In Section 3, we evaluate the models’ performance and explore the importance of feature categories. Section 4 discusses the experimental results, focusing on the impact of various medical factors on cost-effectiveness. Finally, Section 5 concludes the paper with the future scope and limitations of the research work.

## 2. Methodology

### 2.1. Dataset

Our dataset was sourced from Amazon.com (accessed on 8 Januray 2024), where we conducted web crawling in Python 3.10.1 to gather information on four common types of over-the-counter (OTC) medicine: cold, allergy, digestion, and pain relief. These categories were displayed on the website as “Cold & Flu Medicine”, “Allergy Medicine”, “Antacids”, and “Non-aspirin Pain Relievers”, respectively. Notably, each medicine type on Amazon encompassed various subcategories. For instance, additional subcategories existed within the cold medicine category, such as “Cough & Sore Throat Medicine”. To ensure robust data volume for our analysis, we selected the most significant subcategories, such as “Cold & Flu Medicine” and “Antacids” for cold and digestion, respectively. Alternatively, we opted for more diverse subcategories than others within the same medicine type. For example, “Non-aspirin Pain Relievers” was chosen for pain medicine due to its broader range of treatments, despite being the second largest subcategory next to “Joint & Muscle Pain Relief”.

Similarly, “Allergy Medicine” was preferred over “Sinus Medicine” for its potential to provide more general insights into allergy-related symptom treatments. Our dataset comprised 916 records for cold, 618 for allergy, 678 for digestion, and 420 for pain. Each record represented an OTC medicine item available for sale on Amazon, and we collected various information for each medication, as illustrated by the example item in Table 1.

Table 1 demonstrates the detailed information collected for each item through web crawling, exemplified by the DayQuil and NyQuil Combo Pack. This product, designed to provide multi-symptom relief for cold and flu symptoms, including headache, fever, sore throat, and cough, is priced at USD 22.99 and has received high customer ratings. With an average rating of 4.8 stars based on 7081 reviews, most reviewers gave it a 5-star rating (86%). Additionally, the product is eligible for Flexible Spending Account (FSA) or Health Savings Account (HSA) reimbursement and comes in a pack of 72 liquicaps. We also collected information about the product’s dimensions, brand, manufacturer, ingredients, special features, benefits, and usage instructions. Safety information and warnings ensuring safe usage of the product were also included. The ASIN (Amazon Standard Identification Number) and a direct link to the product on Amazon are provided for reference. Subsequently, using this dataset, we developed the “cost-effectiveness rating” (CER) metric for each item by dividing its average rating by its price. In the case of this example item, the CER is calculated as 0.2088 (4.8/22.99). The rest of the attributes in the dataset serve as input factors to analyze their impact on the CER in this research.

### 2.2. Data Preprocessing

To build appropriate machine learning models to identify factors impacting the cost-effectiveness rating of each medicine item, we conducted data preprocessing. Initially, we removed items lacking ratings or prices, as we could not derive the CER for them. This procedure resulted in a reduction of the dataset by 176 records. Additionally, to ensure the robustness of the derived CERs, we eliminated medicine items with fewer than 100 reviews, further reducing our dataset to 1445 records. Upon analyzing the distribution of CERs for each medicine type, we observed that the dataset exhibited right skewness, with some extreme CER values on the right end, as depicted in Figure 2.

Building models directly predicting raw CERs poses challenges in achieving accuracy, and the derived important factors from such models may need to be more reliable [38]. Therefore, we engineered a binary target variable using the median CER of each medicine type as a benchmark. If the CER was above the median, it was assigned a value of 1; otherwise, it was assigned a value of 0. Subsequently, we developed separate machine learning models for each medicine type to predict this binary target variable based on the CER and to identify important factors contributing to cost-effective medicine. For each machine learning model constructed, we employed 5-fold cross-validation to ensure the robustness of the model.

### 2.3. Data Exploration and Feature Engineering

We conducted extensive exploration and feature engineering to develop effective machine learning models for predicting each medicine type’s binary cost-effectiveness ratings (CERs). The following is a comprehensive list of the features developed for each type of medicine, detailed in Table 2. These features are categorized into eight groups: FSA or HSA eligibility, size metrics, brand, manufacturer, active ingredients, special effects, symptom treats, and safety warnings. For text-heavy features such as brand and active ingredients, we counted the frequency of each word, identified keywords, and used one-hot encoding to create corresponding features. For example, for the brand category, we counted the frequency of each brand by medicine type and then converted major brands into binary columns using one-hot encoding. Refer to Section 2.3.1, Section 2.3.2, Section 2.3.3, Section 2.3.4, Section 2.3.5, Section 2.3.6, Section 2.3.7 and Section 2.3.8 for detailed explanations of the features created under each of the eight groups.

#### 2.3.1. FSA or HSA Eligible

This label indicates whether the medicine item is eligible for Flexible Spending Account (FSA) or Health Savings Account (HSA) benefits, indicated as yes or no. Figure 3 depicts the distribution of FSA status across all collected medicine items.

#### 2.3.2. Size

Under this category, we considered three metrics to gauge the size perspective of each item: counts per pack, weight (in ounces), and dimensions (in inches). Each metric was divided into four quantiles (e.g., 1st quantile as the ‘highest quantile’ for size, 2nd quantile as the ‘high quantile’, 3rd as the ‘low quantile’, and 4th as the ‘lowest quantile’). Then, one-hot encoding was utilized to create dummy variables for each quantile. Figure 4 displays the distribution of counts per pack from all collected medicines.

Counts per pack: Refers to the number of items per pack.Weight: Represents the weight of the item.Inches: Indicates the dimensions of the item.

#### 2.3.3. Brand

The label refers to the brand of the item. We counted the frequency of each brand by medicine type, and then, converted major brands into binary columns using one-hot encoding. Figure 5 shows the distribution of the top 10 brands across all collected medicines.

#### 2.3.4. Manufacturer

Similar to the brand feature, we counted the frequency of each manufacturer by medicine type and created individual binary columns for major manufacturers using one-hot encoding. Figure 6 displays the distribution of the top 10 brands among all collected medicines.

#### 2.3.5. Active Ingredients

This label includes the item’s active ingredients. By counting the frequency of each active ingredient by medicine type, we converted the major active ingredients into binary columns using one-hot encoding. Figure 7 demonstrates the distribution of the top 10 ingredients across all collected medicines.

#### 2.3.6. Special Effects

These features indicate whether the medicine item possesses specific properties such as being fast-acting, long-lasting, maximum strength, non-drowsy, and/or kid-friendly. From columns including ‘Product Name,’ ‘Special Feature,’ ‘About,’ and ‘Item Description,’ we identified keywords reflecting each special effect. Figure 8 illustrates the distribution of maximum strength, non-drowsy, kid-friendly, and long-lasting special effects among all the collected medicine items.

#### 2.3.7. Symptom Treats

Symptom treats refer to the number of symptom words each medicine item treats. Major symptom words were identified from columns like ‘Special Benefit’, ‘Special Use’, ‘About’, and ‘Item Description’ based on their frequencies by medicine type. The number of major symptom words associated with each medicine was recorded in a variable. Figure 9 presents the distribution of the top 10 symptom words treated by all collected medicine items.

#### 2.3.8. Safety Warnings

Safety warnings indicate the number of safety concern words associated with each medicine item. Major safety concerns were identified from the ‘Safety Information’ column by counting their frequencies. A variable was created to count each medicine item’s number of major safety concern words. Figure 10 showcases the distribution of the top 10 safety concern words among all the collected medicine items.

### 2.4. Machine Learning Modeling and Impact Assessment of Key Factors

In the previous sections, we established a binary CER target variable based on median values for predicting cost-effectiveness ratings. We engineered features across eight categories, including FSA or HSA eligibility, size metrics, brand, manufacturer, active ingredients, etc. Subsequently, we employed eight machine learning models/classifiers for each medicine type to predict binary CERs, each validated through 5-fold cross-validation. Using various metrics, we identified the optimal machine learning model with the best hyperparameter set for each medicine type. SHAP values were then utilized to evaluate feature importance and identify key categories influencing the CER for each medicine type. Furthermore, logistic regression was employed to determine the direction of impact—whether positive (indicating greater cost-effectiveness) or negative (indicating less cost-effectiveness)—of critical factors.

#### 2.4.1. Machine Learning Models for Predicting CER and Performance Metrics

We utilized eight machine learning models/classifiers to predict binary CERs for each of the four medicine types. These models were chosen based on their proven success in similar applications [25,26,27,28,29,30,31,32,33,34] and their distinct strengths. For models sensitive to data scaling, such as logistic regression, KNN, LDA, Gaussian NB, and MLP, we standardized the features to have a mean of zero and a standard deviation of one, improving convergence and performance. Each model was evaluated through 5-fold cross-validation using the collected medicine dataset and refined using GridSearchCV (GSCV) from the Scikit-Learn library for hyperparameter tuning.

We began with logistic regression because of its straightforwardness and ease of interpretation, using default parameters and a regularization strength (C) of 1.0. Logistic regression’s computational efficiency, with a complexity of O(n), where n is the number of instances, makes it suitable for moderate-sized datasets. Since logistic regression is computationally efficient and has a complexity of O(n), in which n is the number of instances, it is suitable for medium-sized datasets.

Next, we investigated the K-nearest neighbors (KNN) algorithm, chosen for its capability to model non-linear relationships effectively. Different values for the number of neighbors (k) were examined, and an optimal k was selected through GSCV to balance computational efficiency and model accuracy. KNN’s instance-based approach, with a complexity of O(k∙n), is less efficient for large datasets due to the need to compute distances between each pair of instances.

To handle high-dimensional datasets more effectively, we explored tree-based models like the decision tree (DT), adjusting parameters such as maximum depth and minimum sample splits to optimize its performance. Decision trees have a complexity of O(nlogn), making them more efficient than KNN for larger datasets.

Building on decision trees, we implemented random forest (RF), known for robustness and reduced overfitting through ensemble learning. Parameters such as the number of estimators (50, 100, 200, 300), maximum depth (none, 10, 20, 30), and bootstrap options (true, false) were tuned. The computational complexity of random forest is O(m∙nlogn), where m is the number of trees, offering improved robustness at the cost of higher computational demands.

For a more advanced tree-based approach, we employed XGBoost (XGB), a gradient boosting framework. The learning rates (0.01, 0.05, 0.1), number of estimators (100, 200, 300), and maximum depth (3, 5, 7) were tuned. XGBoost leverages regularization to handle missing data and prevent overfitting, with a complexity of O(t∙nlogn), where t is the number of boosting rounds, making it particularly effective for large, high-dimensional datasets.

To enhance our model selection, we incorporated linear discriminant analysis (LDA) and Gaussian naive Bayes (Gaussian NB) due to their probabilistic classification capabilities. LDA optimizes class separation with a complexity of O(n^2^), while Gaussian NB, with a linear complexity of O(n), assumes feature independence and is very efficient.

Lastly, we integrated the multi-layer perceptron classifier (MLP), a neural network model known for its flexibility in capturing complex relationships. The hidden layer sizes (50, 100, 50–50), activation functions (ReLU, tanh), and solvers (Adam, SGD) were tuned, balancing the tradeoff between interpretability and performance. The complexity of MLP is O(e∙n), where e is the number of epochs and n is the number of instances; it has higher computational demands due to its flexibility.

To evaluate the performance of our classifiers, we employed a range of standard metrics, including accuracy, area under the curve (AUC) for the receiver operating characteristic (ROC), precision, recall, and F1-score. Accuracy measures the fraction of correct predictions out of all predictions made. The ROC-AUC curve visually demonstrates the performance of a classifier by plotting recall against the false positive rate at different thresholds. Precision evaluates the proportion of true positive results among all positive predictions, while recall (also known as sensitivity) measures the proportion of true positives identified correctly. The F1-score offers a harmonic mean of precision and recall, delivering a single metric that balances the two.

#### 2.4.2. SHAP Values and Logistic Regression Coefficients for Identifying Factor Impact

Multiple methods are commonly employed to identify critical features within machine learning models. One method entails leveraging the importance of feature from tree-based models, such as decision trees, random forests, and gradient boosting machines. These models calculate feature importance using metrics like Gini impurity or entropy, which assess a feature’s effectiveness in data splitting and uncertainty reduction. Another approach involves examining model coefficients, like those in logistic regression, which indicate the direction and strength of the relationship between features and predicted outcomes. Another approach, permutation importance, assesses performance decreases when feature values are randomly permuted. However, a compelling method is SHAP (Shapley Additive Explanations), which interprets machine learning model outputs by attributing predictions to each feature’s contribution. By employing concepts from cooperative game theory [35], SHAP values provide insights into each feature’s impact on individual predictions. Unlike other methods, SHAP values offer a unified framework for interpreting complex machine learning models, regardless of whether they are logistic regression, tree-based models, or neural networks, making them highly versatile. Additionally, SHAP values inherently account for multicollinearity among features by considering their joint contributions to model predictions [39]. In contrast, tree model feature importance, model coefficients, or permutation importance may overlook multicollinearity issues or interactions between features, potentially leading to biased importance scores.

We determined the magnitude of each factor’s impact by analyzing mean absolute SHAP values. In addition, we developed logistic regression models to investigate the direction of impact (positive or negative coefficients) for each factor concerning each medicine type. While SHAP values can also provide insights into the direction of impact, logistic regression offers a straightforward and intuitive interpretation [35,40]. A positive coefficient for a particular factor indicates that an increase in the feature value leads to a higher likelihood of the positive class, thus being more cost-effective. Conversely, a negative coefficient suggests that an increase in the feature value corresponds to a lower likelihood of the positive class of CER. The direct relationship between the sign of the coefficient and the prediction outcome facilitates a straightforward interpretation of the directional impact of factors on the cost-effectiveness rating. Moreover, compared to SHAP, logistic regression coefficients offer a more straightforward approach for customers to grasp the directional influence of factors, aiding in making more cost-effective purchasing decisions.

## 3. Results

### 3.1. Machine Learning Classifiers for CER Across Medicine Types

In Table 3, Table 4, Table 5 and Table 6, we present a comparison of eight machine learning classifiers across the four types of medicine (cold/allergy/digestion/pain relief), utilizing a 5-fold cross-validation methodology to predict binary cost-effectiveness ratings (CERs). The results are reported as average values with standard deviations. While accuracy and F1 metrics are essential predictive performance indicators, we primarily emphasize the ROC-AUC metric due to its threshold independence and ability to assess the model’s ranking capabilities, which are crucial for correctly identifying true positives, especially in the context of highly cost-effective medications.

Upon examining the results for each medicine type, the choice of the most suitable model varies depending on the specific medication under consideration. The random forest (RF) model emerges as the most effective choice for cold medicine, achieving the highest ROC-AUC of 0.7428 ± 0.0863 among all models, indicating its superior ability to discern between high and low CERs. Despite the simplicity of the logistic regression (LR) model for allergy medicine, it demonstrates robust performance, with an ROC-AUC of 0.7548 ± 0.045, outperforming alternative models such as linear discriminant analysis (LDA). Furthermore, LR exhibits higher average accuracy (0.6793 vs. 0.6480) and average F1-score (0.6849 vs. 0.6394) compared to LDA, which supports its preference. In the case of digestion medicine, random forest (RF) once again showcases its effectiveness with a commendable ROC-AUC of 0.7081 ± 0.071, surpassing logistic regression and XGBoost in predictive capability, as supported by higher accuracy and F1-scores. Lastly, random forest (RF) stands out with the highest ROC-AUC of 0.8022 ± 0.050 for pain relief medicine, underscoring its robust performance and versatility in handling diverse features.

In summary, while random forest (RF) consistently demonstrates commendable performance across different medicine types, the optimal model choice varies based on each medication’s dataset’s unique characteristics and complexities. Consequently, for cold medicine, allergy medicine, digestion medicine, and pain relief medicine, the preferred models are random forest (RF), logistic regression (LR), random forest (RF), and random forest (RF), respectively. Subsequent sections will analyze important features or input factors using the identified best model for each medicine type, assessing their impact on cost-effectiveness ratings.

### 3.2. Key Feature Categories Influencing CER across Medicine Types

Figure 11, Figure 12, Figure 13 and Figure 14 provide insights into the primary factors influencing cost-effectiveness ratings (CERs) across cold, allergy, digestion, and pain relief medicines. These insights are derived from SHAP values calculated using the best model identified for each medicine type. By examining the top five factors’ feature categories in each plot, we discerned the most impactful feature categories for each medicine type from the eight feature categories included in this research (FSA or HSA eligibility, size metrics, brand, manufacturer, active ingredients, special effects, symptom treats, and safety warnings).

In Figure 11, we discern the key feature categories influencing the CERs of cold medicine using the random forest model:FSA or HSA eligibility: This signifies the potential for consumers to utilize pre-tax funds for medication purchases, which is more cost-effective.Symptom treats: The number of symptoms treated emerges as a significant contributor to the CER, underscoring the importance of efficacy considerations.Safety warnings: Safety warnings also significantly contribute to cost-effectiveness ratings, emphasizing the importance of safety considerations.Size metrics: Both lower and higher quantiles of inches play a significant role in influencing the CER, suggesting that the physical dimensions of the medication packaging impact its cost-effectiveness.

In Figure 12, using the logistic regression model, we examine the factors influencing the CERs of allergy medicine and identified key feature categories:Size metrics: Smaller packaging or lighter weight contribute to cost-effectiveness.Manufacturer influence: Specific manufacturers like Johnson & Johnson, Bayer, Sanofi, Major, and Perrigo exert notable influence, indicating that brand reputation and trustworthiness may affect consumer ratings when adjusting the costSpecial effects: Attributes like being kid-friendly influence the CER, enhance safety perceptions, and influence actual cost-effectiveness.Symptom treats: Like cold medicine, the medication’s ability to address a broader range of symptoms impacts the CER.

Figure 13 showcases the factors influencing the CERs of digestion medicine using the random forest model, where we identified vital feature categories:FSA or HSA eligibility: Similar to cold medicine, this suggests the potential for pre-tax fund utilization to be more cost-effective.Size metrics: Similar to allergy medicine, smaller-sized packaging or lighter weight mainly affects actual cost-effectiveness.Symptom treats: Like cold and allergy medicine, the medication’s effectiveness in treating various symptoms influences cost-adjusted ratings.Active ingredients: Specific ingredients like calcium, famotidine, and magnesium influence perceived cost-effectiveness.

Figure 14 uncovers the factors impacting the CERs of pain relief medicine using the random forest model and identifies key feature categories:Size metrics: Both lower and higher quantiles of inches impact packaging dimensions.FSA or HSA eligibility: Signifying potential pre-tax fund usage to be more cost-effective.

These insights offer a comprehensive view of the feature categories driving cost-effectiveness across different medicine types, with further detailed analyses of individual factors from these categories presented in the Discussion section.

## 4. Discussion

Building upon the insights gleaned from the SHAP plots presented in the previous section, which evaluated the relative importance of various factors and identified key feature categories, we developed logistic regression models for each of the four medicine types. By determining the sign of each coefficient, we can distinguish between positive effects (indicating higher cost-effectiveness) and negative effects (indicating lower cost-effectiveness). Figure 15, Figure 16, Figure 17 and Figure 18 illustrate the direction of influence of factors, as determined by logarithmic regression coefficients, with factors across cold, allergy, digestion, and pain relief medicine prominently displayed. 

In Figure 15, we examined the directional impact of key factors for cold medicine. Both ‘FSA or HSA eligible’ and ‘symptom treats count’ showed positive impacts, indicating that medicines eligible for pre-tax funds and those treating more symptoms tend to be more cost-effective typically. Surprisingly, ‘safety warning count’ also positively influenced the CER, suggesting that medicines with safety warnings might offer better cost-effectiveness than those without. When comparing medication with and without safety warnings, we found that those with warnings not only had a lower average price (USD 12.95 vs. USD 19.08) but also received higher average ratings (4.68 vs. 4.57). Further analysis revealed that medicines with safety warnings more frequently contained active ingredients such as dextromethorphan, acetaminophen, and phenylephrine (as shown in Table 7), clinically proven to be effective in treating cold symptoms [41,42,43].

Moreover, Figure 15 highlights phenylephrine and acetaminophen as top factors positively impacting the CER, indicating that including such ingredients contributes to higher ratings for medicines with safety warnings when the price is the same. Therefore, we do not discourage the purchase of cold medicines with safety warnings. They offer cost-effectiveness due to their lower average price and the inclusion of effective ingredients such as phenylephrine and acetaminophen, resulting in higher ratings. However, individuals should consider their allergies before opting for these medicines. Additionally, smaller packaging positively impacts cost-effectiveness, while larger packaging has a negative effect.

In Figure 16, we delved into allergy medicine and uncovered insights into the directional impact of key factors. We found that factors like smaller-sized packaging and lighter weight held positive coefficients, confirming their role in improving cost-effectiveness. Moreover, their positive coefficient indicated that allergy medications featuring kid-friendly special effects demonstrated heightened cost-effectiveness. Similar to cold medicine, allergy remedies addressing a broader array of allergy symptoms generally received higher ratings at comparable prices, thus bolstering cost-effectiveness. When scrutinizing manufacturers, we observed negative coefficients for Johnson & Johnson, Bayer, and Sanofi, while Major and Perrigo exhibited positive coefficients. However, as Table 8 shows, a closer examination of these manufacturers’ average price, rating, and cost-effectiveness ratings (CERs) revealed conflicting outcomes. Despite Perrigo and Major achieving slightly higher ratings, their elevated average prices outweighed the benefits, resulting in lower average CER values, indicating reduced cost-effectiveness. It is possible that interactions between the manufacturer and other feature categories, such as brand, ingredients, and safety warnings, may have influenced the cost-effectiveness ratings, thus altering interpretations of the manufacturer’s logistic coefficient [44,45]. Consequently, relying on manufacturer-based decisions needs more robustness in guiding consumers toward cost-effective allergy medicine purchases. Therefore, in assessing allergy medicine, we primarily focused on other key feature categories identified for the CER, particularly size metrics such as smaller size or lighter weight, special effects, especially those appealing to children; and symptom coverage, particularly medicines capable of addressing a broader range of symptoms.

In Figure 17, we analyzed the directional impact of key factors in digestion medicine. Similar to cold medicine, being FSA or HSA eligible proved more cost-effective, as confirmed by its positive coefficient. Likewise, akin to allergy medicine, smaller-sized packaging or lighter weight also demonstrated increased cost-effectiveness, as indicated by their positive coefficient. Furthermore, like cold and allergy medicine, addressing a broad range of digestion symptoms was more cost-effective. Calcium, famotidine, and magnesium exhibited positive coefficients regarding active ingredients, indicating increased cost-effectiveness. Based on the collected data, top digestion brands containing calcium included Prelief (with 85.71% labeling calcium as an active ingredient), Rolaids, Tums, and Mylanta. For famotidine, Pepcid stood out, with 57.14% labeling famotidine as an active ingredient. Finally, the top digestion brands for magnesium were Rolaids (with 71.43% labeling magnesium as an active ingredient) and Mylanta.

In Figure 18, analyzing pain relief medicine, smaller-sized packaging positively impacts the CER, while larger-sized packaging negatively impacts the CER. Items eligible for FSA or HSA are more cost-effective.

## 5. Conclusions

This study used machine learning to identify key factors influencing the cost-effectiveness of over-the-counter (OTC) medications. The analysis revealed that FSA/HSA eligibility, symptom treatment range, active ingredients, special effects, safety warnings, and packaging size significantly impact cost-effectiveness across cold, allergy, digestion, and pain relief medications. Medications eligible for FSA or HSA funds, those treating a broader range of symptoms, and those with smaller packaging are generally perceived as more cost-effective. For cold medicines, safety warnings maintain cost-effectiveness due to their lower average price and the inclusion of effective ingredients such as phenylephrine and acetaminophen. Allergy medications featuring kid-friendly special effects demonstrated heightened cost-effectiveness. Active ingredients like calcium, famotidine, and magnesium notably boost the cost-effectiveness of digestion medicines. Consumers can use these insights to make more informed choices, ensuring they receive high-quality treatments at optimal prices. For manufacturers and retailers, emphasizing these key factors can improve product appeal and competitiveness. Overall, leveraging machine learning to understand cost-effectiveness helps enhance decision-making for consumers, manufacturers, and retailers in the pharmaceutical industry.

Several limitations should be considered when interpreting the findings. Firstly, the data originate from only four types of medicines available on Amazon Pharmacy, and the sample size is relatively small, which may limit the generalizability of the findings. Additionally, the factors influencing customer decision-making were restricted to the information available on Amazon Pharmacy. The cost-effectiveness of medicines and customer decision-making could be affected by other factors, such as promotional activities and convenience. Furthermore, the self-introduced cost-effectiveness rating (CER) in this study, while intended to represent the perceived value of medicines in terms of user rating and price, does not purely account for the efficacy of the medicines. Instead, it collectively considers medications’ efficacy, quality, and user satisfaction, potentially introducing response bias.

Future research should focus on collecting more extensive and more diverse datasets of medicines to validate these findings across various online medicine platforms. In addition to the machine learning models used in this study, future work should explore additional models, including more advanced techniques like deep neural networks and ensemble methods, to assess whether they offer improved performance or greater accuracy. Moreover, studying the practical implementation of these findings in real-world settings is crucial. Collaboration with online pharmacies and retailers is necessary to incorporate these insights into their recommendation systems and marketing strategies. In this way, we can enhance customer experience, helping them make better-informed decisions based on a comprehensive understanding of cost-effectiveness. Additionally, tracking the outcomes of such implementations will provide valuable feedback for refining these models and approaches, ensuring they remain relevant and effective in practical applications.

## Figures and Tables

**Figure 1 bioengineering-11-00818-f001:**
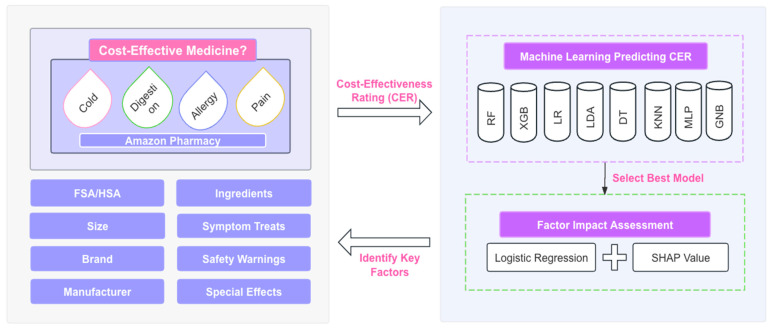
A research framework for identifying key contributors to cost-effective medicines.

**Figure 2 bioengineering-11-00818-f002:**
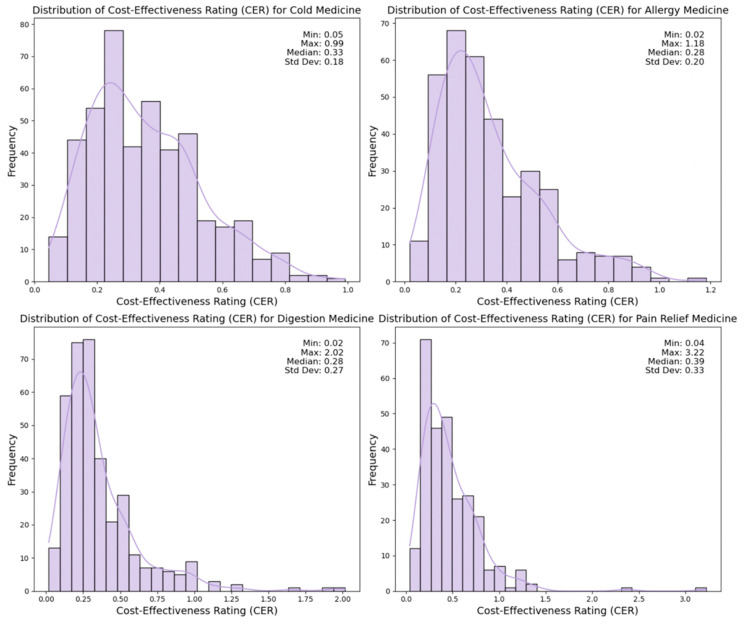
Distribution of cost-effectiveness ratings (CERs) across medicine types.

**Figure 3 bioengineering-11-00818-f003:**
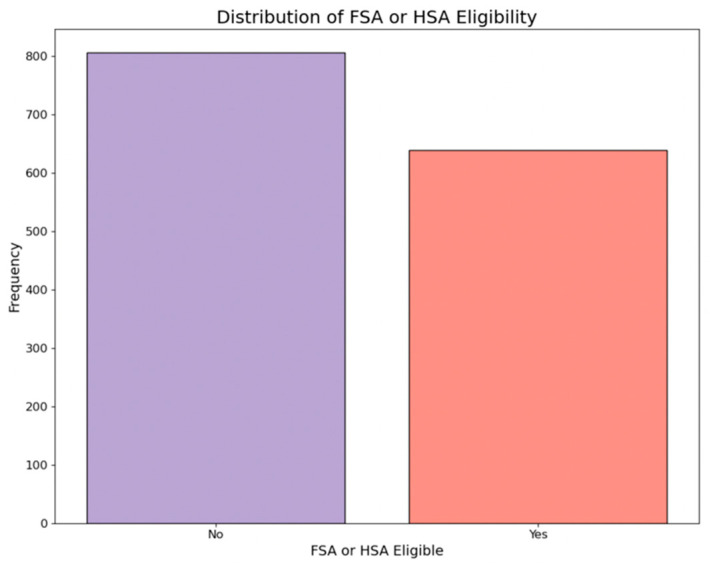
Distribution of FSA or HSA eligibility.

**Figure 4 bioengineering-11-00818-f004:**
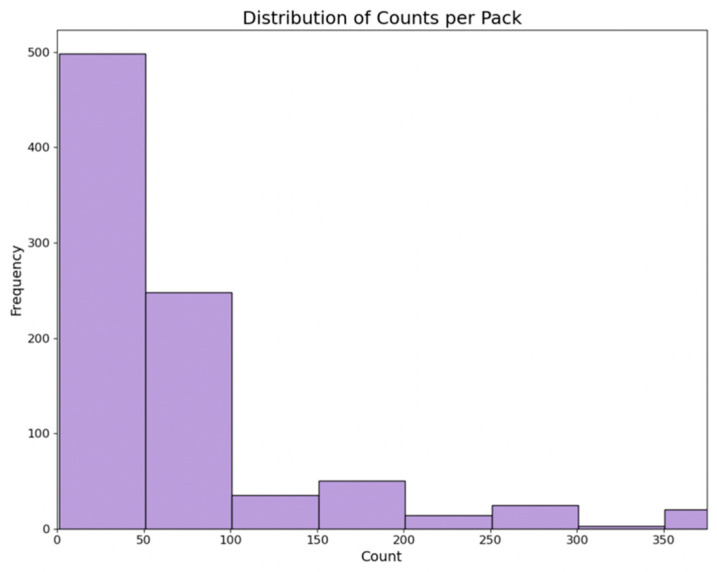
Distribution of counts per pack.

**Figure 5 bioengineering-11-00818-f005:**
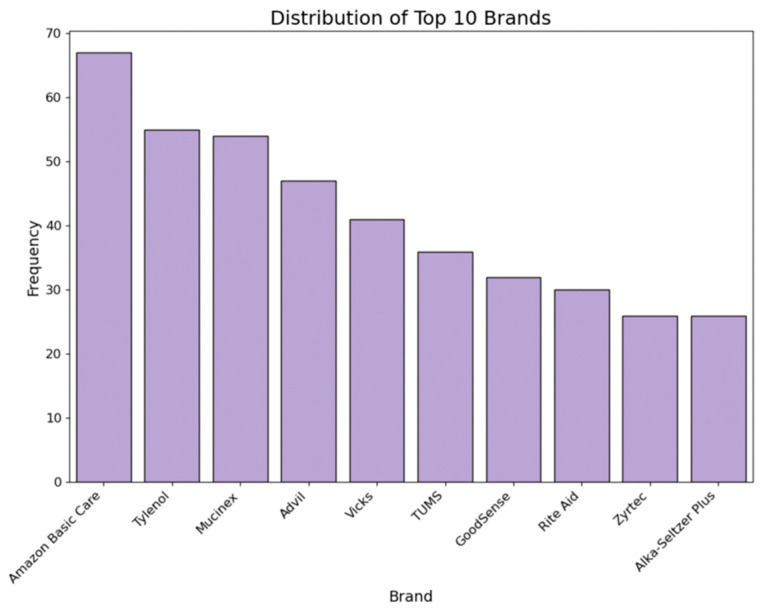
Distribution of top 10 brands.

**Figure 6 bioengineering-11-00818-f006:**
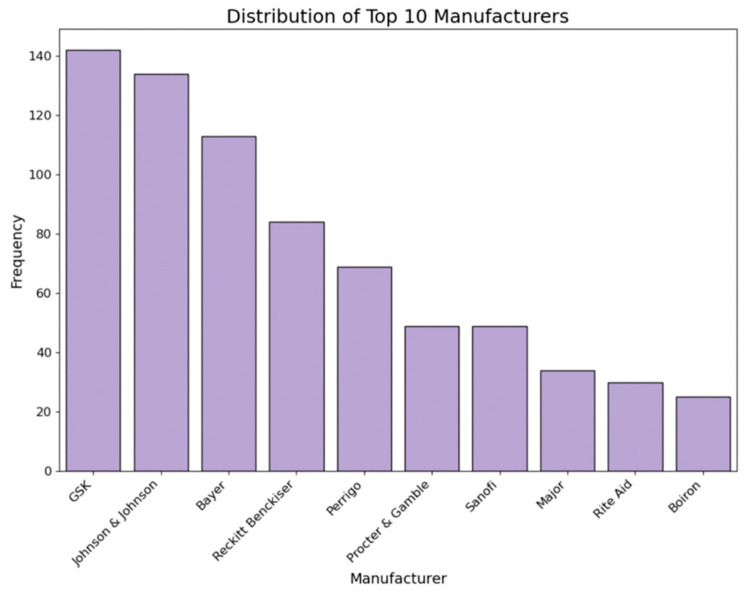
Distribution of top 10 manufacturers.

**Figure 7 bioengineering-11-00818-f007:**
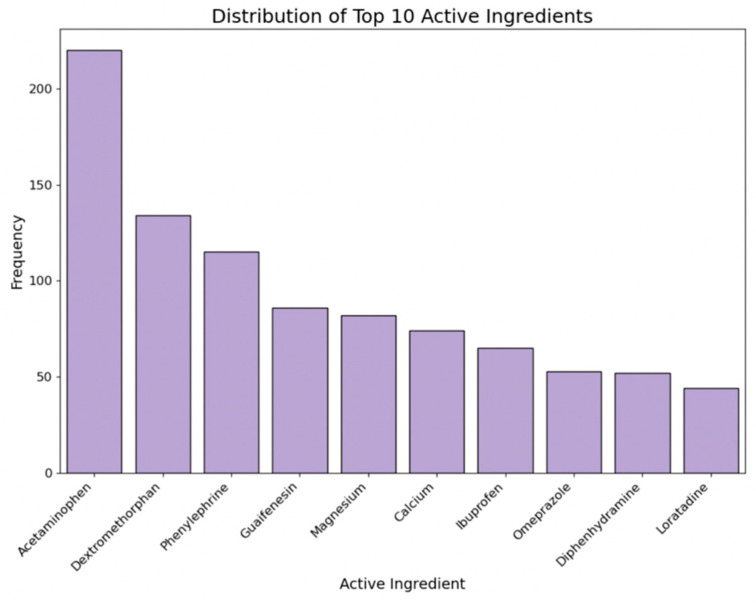
Distribution of top 10 active ingredients.

**Figure 8 bioengineering-11-00818-f008:**
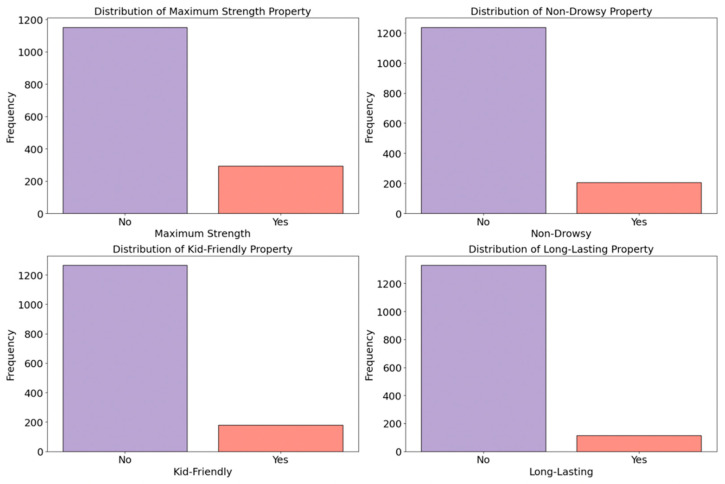
Distribution of special effects (maximum strength/non-drowsy/kid-friendly/long-lasting).

**Figure 9 bioengineering-11-00818-f009:**
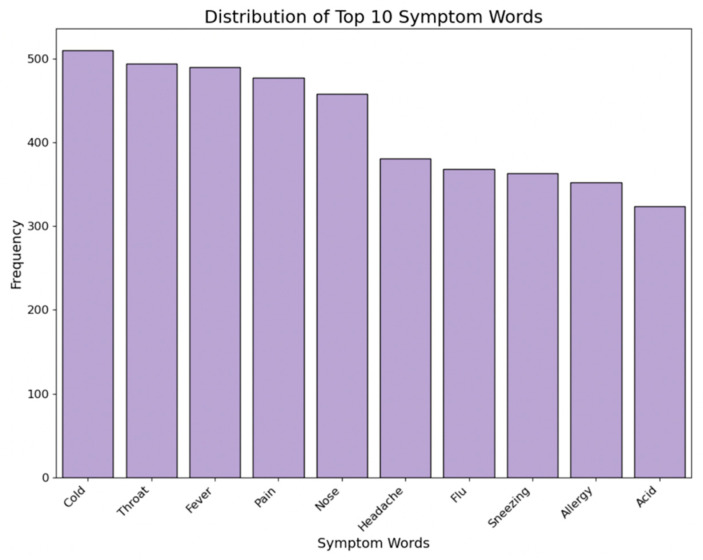
Distribution of symptom words.

**Figure 10 bioengineering-11-00818-f010:**
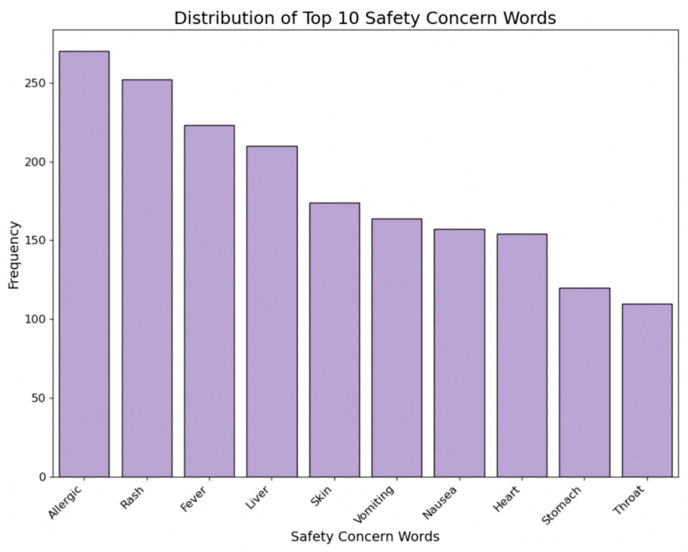
Distribution of safety concern words.

**Figure 11 bioengineering-11-00818-f011:**
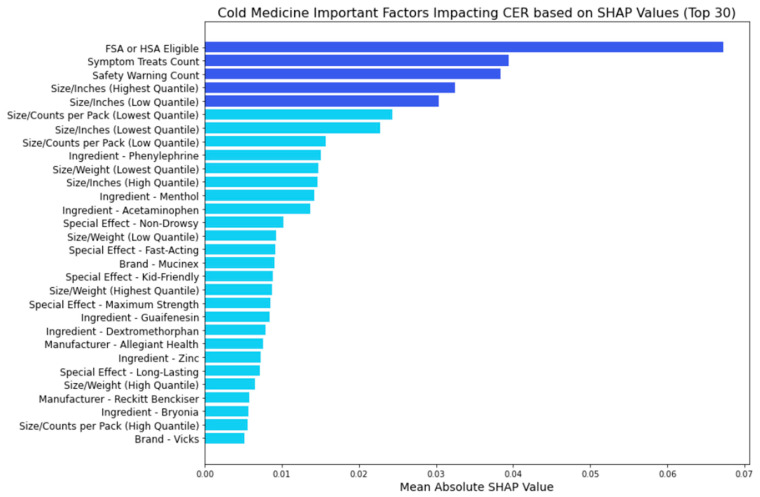
Cold medicine: important factors impacting CER (SHAP).

**Figure 12 bioengineering-11-00818-f012:**
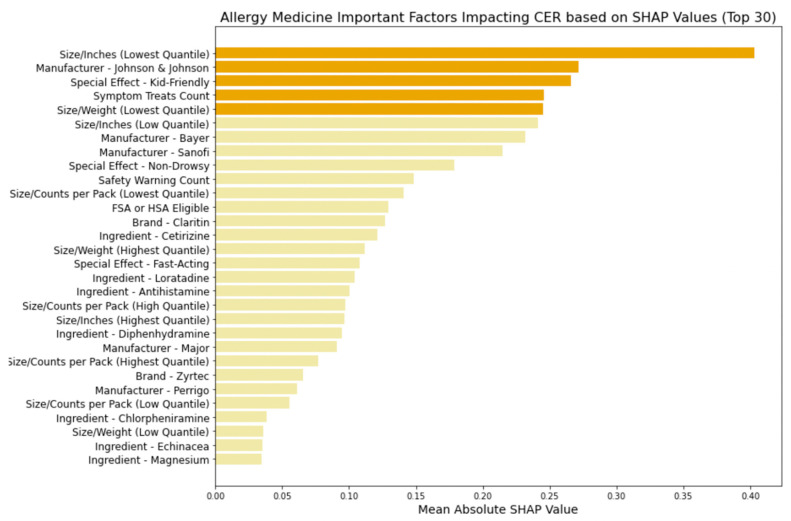
Allergy medicine: important factors impacting CER (SHAP).

**Figure 13 bioengineering-11-00818-f013:**
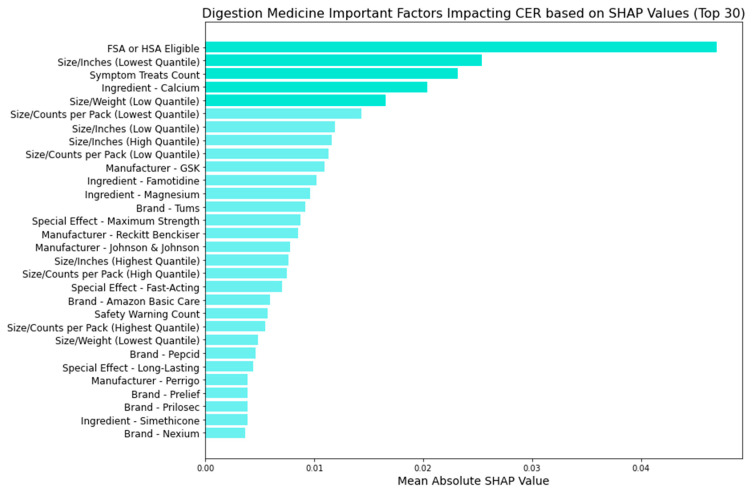
Digestion medicine: important factors impacting CER (SHAP).

**Figure 14 bioengineering-11-00818-f014:**
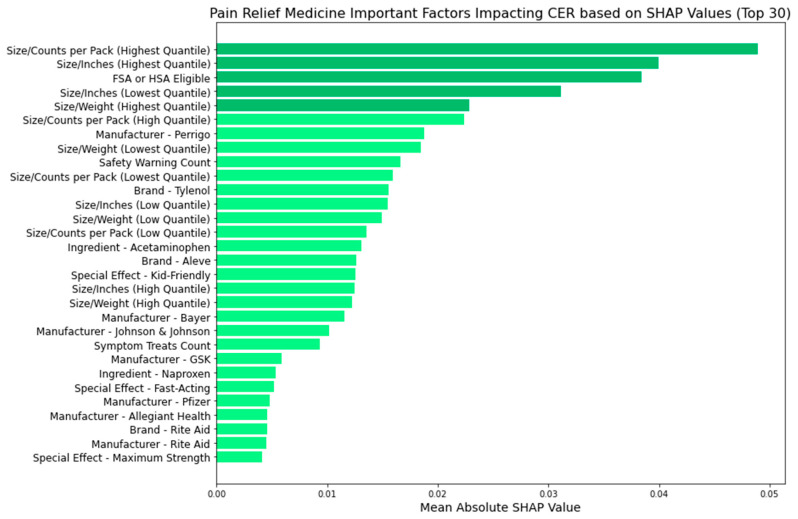
Pain relief medicine: important factors impacting CER (SHAP).

**Figure 15 bioengineering-11-00818-f015:**
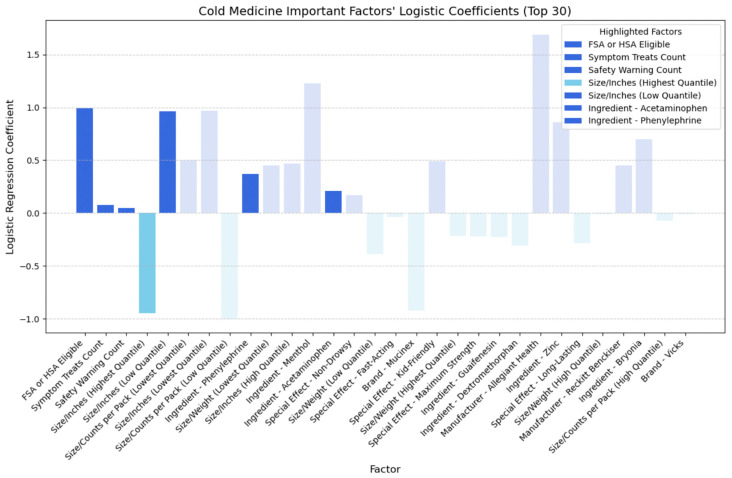
Directional impact of cold medicine factors on CER (logistic regression).

**Figure 16 bioengineering-11-00818-f016:**
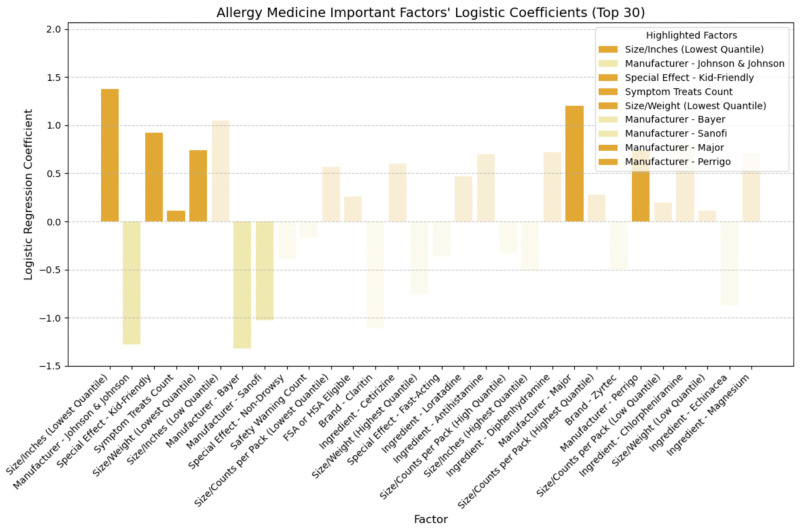
Directional impact of allergy medicine factors on CER (logistic regression).

**Figure 17 bioengineering-11-00818-f017:**
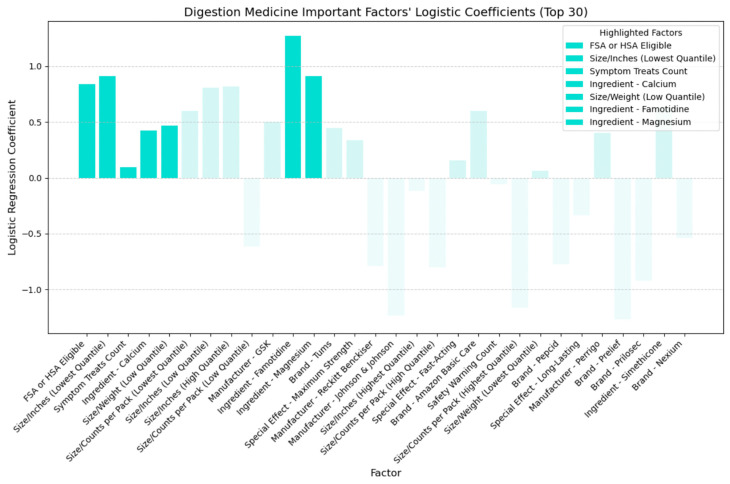
Directional impact of digestion medicine factors on CER (logistic regression).

**Figure 18 bioengineering-11-00818-f018:**
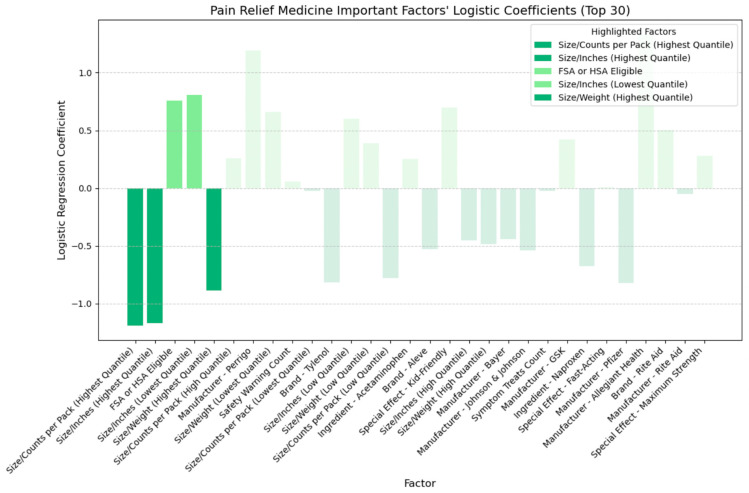
Directional impact of pain relief medicine factors on CER (logistic regression).

**Table 1 bioengineering-11-00818-t001:** Web-crawled data for medicine items is illustrated with an example.

Column	Value
Product Name	DayQuil and NyQuil Combo Pack, Cold & Flu Medicine, Powerful Multi-Symptom Daytime And Nighttime Relief For Headache, Fever, Sore Throat, Cough, 72 Count, 48 DayQuil, 24 NyQuil Liquicaps
Price	USD 22.99
Rating	4.80
Number of Reviews	7081
% 5 Star Reviews	86%
% 4 Star Reviews	10%
% 3 Star Reviews	3%
% 2 Star Reviews	1%
% 1 Star Reviews	1%
Size	72 count (pack of 1)
Item Weight	0.01 ounces
Item Dimension	4.38 × 3 × 3.38 inches
Product Dimension	4.38 × 3 × 3.38 inches; 0.01 ounces
FSA or HSA Eligible	Yes
Brand	Vicks
Manufacturer	Procter & Gamble—HABA Hub
Ingredients	DayQuil Cold & Flu Active Ingredients (In Each Liquicap): Acetaminophen 325 mg (Pain Reliever/Fever Reducer), Dextromethorphan HBr 10 mg (Cough Suppressant), Phenylephrine HCl 5 mg (Nasal Decongestant) Inactive Ingredients: FD&C Red No. 40, FD&C Yellow No. 6, Gelatin, … *(See full list in original text)*
Special Feature	Non-drowsy
Product Benefit	Cough, Cold & Flu Relief, Sore Throat, Fever, & Congestion Relief
Special Use	Cold, Cough, Sore Throat, Fever
About	About this item—FAST, POWERFUL MULTI-SYMPTOM RELIEF: Use non-drowsy DayQuil for daytime relief and at night try NyQuil for fast relief so you can rest EFFECTIVE COLD & FLU SYMPTOM RELIEF: DayQuil and NyQuil Cold & Flu medicine temporarily relieve common cold & flu symptoms FEEL BETTER FAST: Just one dose starts working fast… *(See full description in original text)*
Item Description	Knock your cold out with Vicks DayQuil and NyQuil SEVERE Cold & Flu Liquid medicine. Just one dose starts working fast to relieve 9 of your worst cold and flu symptoms, to help take you from 9 to none. From the world’s #1 selling OTC cough and cold brand, Vicks DayQuil and NyQuil SEVERE provide fast, powerful, maximum strength relief… *(See full description in original text)*
Safety Information	Safety Information DayQuil Cold & Flu: Liver warning: This product contains acetaminophen. Severe liver damage may occur if you take: • More than 4 doses in 24 h, which is the maximum daily amount for this product • Other drugs containing acetaminophen • 3 or more alcoholic drinks every day while using this product. Sore throat warning: If sore throat is severe… *(see full safety information in original text)*
Directions	Take only as directed—see Overdose warning. Do not exceed 4 doses per 24 h. Adults and children 12 years and over: 2 LiquiCaps with water every 4 h… *(See full directions in original text)*
ASIN	B00796NI1Q
Link	https://www.amazon.com/Vicks-Medicine-Multi-Symptom-Nighttime-Liquicaps/dp/B00796NI1Q/ref=sr_1_22?c=ts&keywords=Cold+%26+Flu+Medicine&qid=1699298540&refinements=p_85%3A2470955011&refresh=1&rps=1&s=hpc&sr=1-22&ts_id=3761171 (accessed on 8 January 2024).

**Table 2 bioengineering-11-00818-t002:** Overview of features.

Feature Category	Feature	Explanation	Feature Type
FSA or HSA Eligible	FSA or HSA Eligible	Indicates if the medicine item is a Flexible Spending Account (FSA) or Health Savings Account (HSA) eligible item (yes/no)	Binary
Size	Counts per Pack	Indicates if the counts per pack belong to the lowest/low/high/highest quantile	Binary
	Weight	Indicates if the weight of the item (in ounces) belongs to the lowest/low/high/highest quantile	Binary
	Inches	Indicates if the dimensions of the item (in inches) belong to the lowest/low/high/highest quantile	Binary
Brand	Brand	Indicates the brand of the item (yes for corresponding one-hot-encoded brand column, no for others)	Binary
Manufacturer	Manufacturer	Indicates the manufacturer of the item (yes for corresponding one-hot-encoded manufacturer column, no for others)	Binary
Ingredients	Active Ingredients	Indicates the presence of active ingredients (yes for corresponding one-hot-encoded ingredient columns, no if an ingredient is absent)	Binary
Special Effect	Fast-Acting	Indicates if the item qualifies as a fast-acting property	Binary
	Long-Lasting	Indicates if the item qualifies as a long-lasting property	Binary
	Maximum Strength	Indicates if the item has maximum strength property	Binary
	Non-Drowsy	Indicates if the item qualifies as non-drowsy property	Binary
	Kid-Friendly	Indicates if the item qualifies as a kid-friendly property	Binary
Symptom Treats	Symptom Treats Count	Number of symptom words this medicine item treats	Numerical
Safety Warnings	Safety Warning Count	Number of safety concern words this medicine item has	Numerical

**Table 3 bioengineering-11-00818-t003:** Cross-validation results of cold medicine machine learning classifier.

	ROC-AUC	Accuracy	Precision	Recall	F1-Score
Random Forest	0.7428 ± 0.0863	0.6897 ± 0.0743	0.7076 ± 0.0914	0.6667 ± 0.1849	0.6703 ± 0.1142
XGBoost	0.7256 ± 0.0886	0.6853 ± 0.0723	0.7026 ± 0.0797	0.6533 ± 0.1798	0.6619 ± 0.1186
Logistic Regression	0.7064 ± 0.0867	0.6364 ± 0.0844	0.6386 ± 0.1037	0.6311 ± 0.2092	0.6188 ± 0.1320
Linear Discriminant Analysis	0.7030 ± 0.0831	0.6187 ± 0.0674	0.6151 ± 0.0613	0.6178 ± 0.1888	0.6046 ± 0.1108
Multi-Layer Perceptron	0.6843 ± 0.0650	0.6322 ± 0.0825	0.6304 ± 0.0790	0.7200 ± 0.1719	0.6560 ± 0.0791
Gaussian Naïve Bayes	0.6473 ± 0.0483	0.5675 ± 0.0568	0.6456 ± 0.1173	0.2844 ± 0.1074	0.3880 ± 0.1127
K-Nearest Neighbors	0.6351 ± 0.0612	0.5944 ± 0.0702	0.6276 ± 0.1004	0.5422 ± 0.2064	0.5541 ± 0.1196
Decision Tree	0.6252 ± 0.0527	0.6322 ± 0.0557	0.6386 ± 0.1037	0.6311 ± 0.2092	0.6188 ± 0.1320

**Table 4 bioengineering-11-00818-t004:** Cross-validation results of allergy medicine machine learning classifier.

	ROC-AUC	Accuracy	Precision	Recall	F1-Score
Logistic Regression	0.7548 ± 0.045	0.6793 ± 0.054	0.6859 ± 0.082	0.6997 ± 0.099	0.6849 ± 0.049
Linear Discriminant Analysis	0.7449 ± 0.044	0.6480 ± 0.050	0.6630 ± 0.070	0.6373 ± 0.131	0.6394 ± 0.065
Multi-Layer Perceptron	0.7269 ± 0.023	0.6734 ± 0.038	0.6569 ± 0.030	0.7278 ± 0.086	0.6884 ± 0.045
Random Forest	0.7223 ± 0.037	0.6736 ± 0.053	0.6730 ± 0.068	0.6994 ± 0.064	0.6823 ± 0.043
XGBoost	0.7160 ± 0.054	0.6679 ± 0.051	0.6780 ± 0.068	0.6598 ± 0.084	0.6641 ± 0.053
Gaussian Naïve Bayes	0.7158 ± 0.013	0.5738 ± 0.044	0.7340 ± 0.179	0.2503 ± 0.103	0.3596 ± 0.109
Decision Tree	0.6131 ± 0.058	0.6137 ± 0.053	0.6219 ± 0.056	0.5798 ± 0.077	0.5988 ± 0.061
K-Nearest Neighbors	0.6044 ± 0.069	0.5828 ± 0.066	0.6153 ± 0.105	0.5002 ± 0.068	0.5454 ± 0.056

**Table 5 bioengineering-11-00818-t005:** Cross-validation results of digestion medicine machine learning classifier.

	ROC-AUC	Accuracy	Precision	Recall	F1-Score
Random Forest	0.7081 ± 0.071	0.6641 ± 0.035	0.7008 ± 0.075	0.6323 ± 0.155	0.6455 ± 0.058
XGBoost	0.7023 ± 0.046	0.6587 ± 0.045	0.6848 ± 0.082	0.6547 ± 0.125	0.6535 ± 0.044
Logistic Regression	0.7004 ± 0.062	0.6150 ± 0.059	0.6254 ± 0.069	0.6335 ± 0.063	0.6233 ± 0.022
Linear Discriminant Analysis	0.6777 ± 0.070	0.6178 ± 0.076	0.6220 ± 0.076	0.6505 ± 0.034	0.6328 ± 0.044
Gaussian Naïve Bayes	0.6410 ± 0.027	0.5494 ± 0.051	0.5410 ± 0.048	0.8243 ± 0.109	0.6455 ± 0.020
K-Nearest Neighbors	0.6351 ± 0.088	0.5604 ± 0.074	0.6031 ± 0.105	0.3974 ± 0.115	0.4680 ± 0.102
Multi-Layer Perceptron	0.6351 ± 0.088	0.6148 ± 0.054	0.5773 ± 0.036	0.8743 ± 0.051	0.6947 ± 0.036
Decision Tree	0.6018 ± 0.059	0.5986 ± 0.052	0.6030 ± 0.060	0.6114 ± 0.049	0.6043 ± 0.037

**Table 6 bioengineering-11-00818-t006:** Cross-validation results of pain relief medicine machine learning classifier.

	ROC-AUC	Accuracy	Precision	Recall	F1-Score
Random Forest	0.8022 ± 0.050	0.7576 ± 0.055	0.7748 ± 0.072	0.7185 ± 0.069	0.7433 ± 0.056
Linear Discriminant Analysis	0.7884 ± 0.063	0.7432 ± 0.066	0.7543 ± 0.093	0.7259 ± 0.050	0.7363 ± 0.055
Logistic Regression	0.7874 ± 0.064	0.7179 ± 0.076	0.7326 ± 0.098	0.6889 ± 0.055	0.7070 ± 0.065
Gaussian Naïve Bayes	0.7867 ± 0.061	0.6594 ± 0.082	0.8042 ± 0.148	0.3852 ± 0.127	0.5168 ± 0.145
XGBoost	0.7577 ± 0.055	0.7286 ± 0.058	0.7240 ± 0.066	0.7259 ± 0.065	0.7235 ± 0.057
Multi-Layer Perceptron	0.7139 ± 0.091	0.6598 ± 0.084	0.6337 ± 0.072	0.7407 ± 0.105	0.6798 ± 0.075
K-Nearest Neighbors	0.6542 ± 0.030	0.5869 ± 0.024	0.5817 ± 0.027	0.5556 ± 0.081	0.5652 ± 0.049
Decision Tree	0.6373 ± 0.039	0.6378 ± 0.040	0.6450 ± 0.065	0.6000 ± 0.049	0.6186 ± 0.034

**Table 7 bioengineering-11-00818-t007:** Chi-square test results for statistically significant active ingredient percentage difference in cold medicines with and without safety warnings (*p*-value < 0.05).

Active Ingredient	Chi-Square Statistic	*p*-Value	Item Count
Dextromethorphan	41.3911	1.25 × 10^−10^	131
Acetaminophen	40.7375	1.74 × 10^−10^	112
Phenylephrine	35.3099	2.81 × 10^−9^	106
Guaifenesin	5.9919	1.44 × 10^−2^	85
Doxylamine	39.091	4.05 × 10^−10^	40
Hydrobromide	17.634	2.68 × 10^−5^	32
Bryonia	5.4334	1.98 × 10^−2^	23
Phosphorus	3.9605	4.66 × 10^−2^	17
Gelsemium	5.9838	1.44 × 10^−2^	15
Ipecacuanha	4.4107	3.57 × 10^−2^	14
Eupatorium	8.8677	2.90 × 10^−3^	13
Perfoliatum	6.9362	8.45 × 10^−3^	12

**Table 8 bioengineering-11-00818-t008:** Comparison of manufacturer-based cost-effectiveness for allergy medicine.

Manufacturer	Average Price	Average Rating	Average CER
Johnson & Johnson	USD 12.4	4.68	0.47
Bayer	USD 19.68	4.57	0.35
Sanofi	USD 11.74	4.71	0.51
Major	USD 25.66	4.72	0.22
Perrigo	USD 22.55	4.73	0.3

## Data Availability

The original data presented in this study are openly available in FigShare at https://doi.org/10.6084/m9.figshare.26390803 (accessed on 8 January 2024).

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
