# Peer review of "Deciphering Factors Contributing to Cost-Effective Medicine Using Machine Learning"

_bioengineering, 2024, doi:10.3390/bioengineering11080818_

Round 1

Reviewer 1 Report

Comments and Suggestions for Authors

The header and footer used 2023, was this paper submitted in 2023 ?

 The paper is about over the counter products, but the data are from Amazon.

Is Amazon the national standard for about over the counter products for each country in the whole world?

In the Abstract, FSA/HAS was not explained. The acronym should be explained at first use.

In the abstract, “assist manufacturers and retailers in enhancing product competitiveness”, but from the content of the paper, there is no formal proof to support such a claim.

In section one, several textbook basic machine learning methods were mentioned. Explain the reasons to use textbook basic machine learning methods instead of state of art latest machine learning methods.

In section one, accuracy is selected as evaluation criteria. Explain why accuracy is important for this application comparing with other evaluation criteria.

In section one, learning method (such as LDA)  is selected. Explain why the solution space is linear.

Section 2.1 explained dataset, but the paper did not provide download link of data for readers.

In Table 1, URL of the product at Amazon is even listed as one factor, how is this important for bioengineering research?

In Table 1, data are text from amazon web site. Naturally, NLP, text analysis, medicine and healthcare domain knowledge would be expected for this research. But this paper did not explain such details.

Section 2.1 discussed number of records. Have the data sets been standardised or normalised?

In section 2.3.3, binary is used before Figure 4. Explain why the solution is binary.

In section 2.3.4, binary is used before Figure 5. Explain why the solution is binary.

In section 2.3.5, binary is used before Figure 6. Explain why the solution is binary.

In section 3.1, binary is used. Explain why the solution is binary.

In Figure 10, are listed factors independent from each other? Such as how is size / inches (highest quantile) independent from size / inches (high quantile)?

In Figure 11, are listed factors independent from each other? Such as how is size / inches (lowest quantile) independent from size / inches (low quantile)?

In Figure 12, are listed factors independent from each other? Such as, how is size / inches (lowest quantile) independent from size / inches (low quantile)?

In Figure 13, are listed factors independent from each other? Such as, how is size / inches (lowest quantile) independent from size / inches (low quantile)?

In Figure 14, are listed factors independent from each other? Such as, how is size / inches (lowest quantile) independent from size / inches (low quantile)?

In Figure 15, are listed factors independent from each other? Such as, how is size / inches (lowest quantile) independent from size / inches (low quantile)?

In Figure 16, are listed factors independent from each other? Such as, how is size / counts per pack (lowest quantile) independent from size / counts per pack (low quantile)?

In Figure 17, are listed factors independent from each other? Such as, how is size / counts per pack (highest quantile) independent from size / counts per pack (high quantile)?

For 2024 references, there is only one with access time in 2024 and it is not clear about the publishing year of the reference item. Please provide publishing time for reference item 1.

There is only one 2023 reference item.

There are two 2022 reference items.

The paper should discuss more relevant recent work, such as:

A Hierarchical Method for Locating the Interferometric Fringes of Celestial Sources in the Visibility Data

Research in Astronomy and Astrophysics 24 (3), 035011, 2024

The journal is titled bioengineering.

Explain how is this paper important for bioengineering.

The title used “Machine Learning”. Explain what is own innovation about what machine learning specific technology.

After Tables and Figures, please explain more details of the results.

The draft may explain more on own original research, such as on data analysis methods.

From computing point of view, the paper should explain in good technical details on computing and space complexity of the computing solutions, such as compare own solution with existing solutions.

The paper should explain from computing perspective how the solutions select parameters and how initial values of parameters were assigned.

The paper reported some numeric results, such as in Tables and Figures. From a data science point of view, authors should explain the generalisability of such results. Authors should also explain limitations, such as possible bias in sampling or data collection.

Comments on the Quality of English Language

Quality of English Language can be improved, such as to use more specific accurate terms.

Author Response

Dear Reviewer,

Thank you for the invaluable suggestions and feedback. We have thoroughly considered them and revised our manuscript accordingly. Please see the responses to each point in red in the attached Word document.

Best,

Bowen

Reviewer 2 Report

Comments and Suggestions for Authors

After examining the article. There are a few major suggestions: 

1. More thorough details on the machine learning models that were used, including the precise techniques and parameters, should be included in the methodology section. The use of logistic regression and machine learning is mentioned in the text, but there are no in-depth explanations of the models or their setups.
2. The dataset is said to be accessible upon request from the authors of the study. It would be advantageous to include a direct link to the dataset or include it as supplemental material for improved repeatability and transparency.

2. To support the conclusions, the results section should include more thorough tables and figures. For example, while eight machine learning classifiers are evaluated in the publication, specific performance measures for each classifier are not provided. It would strengthen the conclusions' robustness to include these facts.

3. Although noted, the examination of manufacturers' effects on cost-effectiveness evaluations is not fully discussed. It would be helpful to have a more thorough explanation of the evaluation methods used for the various manufacturers' goods as well as the ramifications of the results.
4. Based on the study's results, the conclusion has to be revised to provide more detailed suggestions for customers, producers, and merchants. This would provide useful information and actionable conclusions from the study.

5. Make sure that every table and figure has a clear caption and is cited within the text. Table 8 and Figure 15, for instance, are addressed, but their significance and context have to be more clearly included in the conversation.
6. Please check each reference of the article carefully. Various items are missing in the references. 

Author Response

(The authors gave the same response as above.)

Reviewer 3 Report

Comments and Suggestions for Authors

The paper studies factors influencing the cost-effectiveness of over-the-counter (OTC) medications. The dataset was collected via Amazon web crawls for cold, allergy, digestion, and pain relief purposes. A cost-effectiveness rating (CER) metric was developed to score medications based on user ratings and prices. The authors used various ML models to predict the binary CER and identify important factors. The paper later used SHAP values and logistic regression to determine the direction and strength of key factors. The study is complete and meaningful. I thus recommend it to be published after minor corrections of the following points.

[Minor issues]

1. The limitation of the study is not discussed in the manuscript:

i) The authors should discuss the data collection bias as they rely solely on Amazon data. 

ii) The authors should discuss the circumstances where their new CER metric might not capture the true effectiveness of a medication. 

2. In Table 8, the unit of "average price" is missing.

[Suggestion]

I recommend the authors publish or store their data in an open-access repository for follow-up studies by others.

Author Response

(The authors gave the same response as above.)

Reviewer 4 Report

Comments and Suggestions for Authors

The manuscript is devoted to a development of cost-effectiveness indicators for medical drugs based on the machine learning approaches. 

There are the following comments:

1. The manuscript submitted to the Section “Biosignal Processing”, Special Issue “Deciphering Medicine: The Role of Explainable Artificial Intelligence in Healthcare Innovations”. The authors need to explain the relevance, since the paper does not present anything about the explainability of AI. The correspondence of the manuscript to the aims and scope of the Bioengineering journal should be commented on too.

2. What is the fundamental scientific novelty of this article? The authors consider a local applied problem of identifying relationships in a specific dataset using standard machine learning methods.

3. The authors should more clearly present the point-by-point contribution of this research to the subject area.

4. At the end of Introduction, it is necessary to add a description of the further structure of the manuscript.

5. The quality of the figures in the manuscript should be improved. Currently there is significant pixelation.

6. There is no Section 5 in the manuscript, only Sections 4 and 6.

Author Response

(The authors gave the same response as above.)

Round 2

Reviewer 1 Report

Comments and Suggestions for Authors

The revision is light English editing, hence the previous concerns about the quality remain.

Comments on the Quality of English Language

The revision is improved in English.

Author Response

Thank you for the invaluable suggestions and feedback. We have thoroughly considered them and revised our manuscript accordingly. Please see the responses as attached.

Reviewer 2 Report

Comments and Suggestions for Authors

The article has 15% plagiarism and the first source is 5%,which is very high.

The authors have to reduce plagiarism and no source should be greater than 1%

Author Response

(The authors gave the same response as above.)

Reviewer 4 Report

Comments and Suggestions for Authors

Responses seem clear, but:

1. The relevance to the journal as well as contributions should be presented in the manuscript. Lines 183–205 do not contain it.

2. There are no manuscript structures in lines 205-228.

3. Figures and corresponding captions should not be on different pages (see, for example, Figure 16).

4. There is no line 908 (see your response 6).

5. The text and the comments should be improved significantly.

Author Response

(The authors gave the same response as above.)
